# Resveratrol, ε-Viniferin, and Vitisin B from Vine: Comparison of Their In Vitro Antioxidant Activities and Study of Their Interactions

**DOI:** 10.3390/molecules28227521

**Published:** 2023-11-10

**Authors:** Biranty Sy, Stéphanie Krisa, Tristan Richard, Arnaud Courtois

**Affiliations:** 1Université de Bordeaux, Institute of Vine and Wine Sciences, INRAE, Bordeaux INP, Bordeaux Sciences Agro, OENO, UMR 1366, ISVV, 33140 Villenave d’Ornon, France; biranty.sy@u-bordeaux.fr (B.S.); stephanie.krisa@u-bordeaux.fr (S.K.); 2CHU de Bordeaux, Centre Antipoison de Nouvelle Aquitaine, Emergency Building, 33076 Bordeaux, France

**Keywords:** stilbenes, resveratrol, ε-viniferin, vitisin B, interactions, antioxidant

## Abstract

The control of oxidative stress with natural active substances could limit the development of numerous pathologies. Our objective was to study the antiradical effects of resveratrol (RSV), ε-viniferin (VNF), and vitisin B (VB) alone or in combination, and those of a standardized stilbene-enriched vine extract (SSVE). In the DPPH-, FRAP-, and NO-scavenging assays, RSV presented the highest activity with an IC_50_ of 81.92 ± 9.17, 13.36 ± 0.91, and 200.68 ± 15.40 µM, respectively. All binary combinations resulted in additive interactions in the DPPH- and NO-scavenging assays. In the FRAP assay, a synergic interaction for RSV + VNF, an additive for VNF + VB, and an antagonistic for RSV + VB were observed. The ternary combination of RSV + VNF + VB elicited an additive interaction in the DPPH assay and a synergic interaction in the FRAP- and NO-scavenging assays. There was no significant difference between the antioxidant activity of the SSVE and that of the combination of RSV + VNF. In conclusion, RSV presented the highest effects, followed by VNF and VB. The interactions revealed additive or synergistic effects, depending on the combination of the stilbenes and assay.

## 1. Introduction

Reactive oxygen species (ROS) and reactive nitrogen species (RNS) are co-products of normal cellular metabolism. Some of these species play an important role in cell signaling, differentiation, survival, and death. These reactive species can exist in radical forms, containing one or more unpaired, non-radical electron. ROS include superoxide anions (O2^•−^), hydroxyl radicals (^•^OH), hydrogen peroxide (H_2_O_2_), and hypochlorous acid (HClO); RNS include nitric oxide (NO) and peroxynitrite (ONOO^−^) [1].

When these radicals are produced in excessive quantities, they can lead to oxidative or nitrosative stress. Oxidative stress is a physiological condition that occurs when the body’s antioxidant defense systems (enzymatic or non-enzymatic) lose their ability to neutralize excesses of reactive oxygen and nitrogen species, leading to the oxidation of biological macromolecules such as nucleic acids, proteins, and lipids. Numerous studies have shown that they are involved in the pathophysiology of numerous chronic diseases such as cardiovascular, inflammatory, metabolic, and neurodegenerative diseases, and especially cancers [2].

In addition to endogenous antioxidant defense systems, protection against ROS/RNS involves exogenous antioxidants capable of preventing their formation or promoting their elimination. Plant-based foods and beverages are the main sources of antioxidants such as vitamins and phenolic phytochemicals. Dietary polyphenols are the most abundant antioxidants in our diet. The antioxidant effects of polyphenols are due to their reducing power, by donating a hydrogen atom to a wide range of ROS or by scavenging them. They also have the ability to chelate transition metals (Fe and Cu), thereby directly reducing the Fenton reaction and preventing oxidation caused by highly reactive hydroxyl radicals [3,4].

Numerous methods measure antioxidant capacity. Those methods are based on the scavenging or reduction of free and stable radicals and are convenient to identify the various antioxidant mechanisms existing from one phenolic compound to another. These assays include the scavenging of NO, the reduction of ABTS, DPPH, or peroxide radicals in the ORAC methods, and the reduction of a ferric derivative to a ferrous iron derivative in the ferric reducing antioxidant power (FRAP) assay. In studies that aimed to understand the nature of the antioxidant activities of natural compounds, a multi-method approach has been used to evaluate the different mechanisms of action of antioxidants [5].

In food or nutraceuticals, compounds are present in complex mixtures and can therefore interact with each other. The resulting biological effect may then be the result of additive, synergistic, or antagonistic interactions. Three types of interactions can occur: additivity, synergism, or antagonism. Additivity occurs when the combination of two or more molecules gives an effect identical to the sum of the effects of the individual molecules, synergism when the effect of the combination is greater than the expected effect of the individual molecules, and antagonism when the effect of the combination is less than the expected effect of the individual molecules. Antioxidants can synergistically interact through a regenerative mechanism, i.e., one antioxidant regenerates the other. With regard to polyphenols, Aftab and Vieira highlighted a mechanism involving resveratrol in the regeneration of the reduced form of curcumin. Resveratrol was able to regenerate oxidized curcumin, thereby increasing the antioxidant activity of curcumin [6].

Stilbenes are polyphenols known for their antioxidant and anti-inflammatory activities [7,8,9,10]. The main compound of this family is resveratrol, the oligomerization of which can produce numerous stilbenes containing up to eight resveratrol units. Numerous studies have shown their benefits, especially for resveratrol, in the prevention of diseases that involved inflammation and oxidative stress in their physiopathology. ε-Viniferin (a resveratrol dimer) was also shown to exhibit antioxidant and anti-inflammatory, but also anti-proliferative, neuroprotective, and anti-adipogenic properties [7]. In contrast, vitisin B (a resveratrol tetramer) has been poorly studied, but some results have suggested that this compound could exhibit similar properties or even be more active than resveratrol [8]. In most studies on their biological activities, these compounds are individually used. However, in plants, they are present in mixtures in variable amounts and proportions, and their interactions have been poorly studied. The aim of our study was to individually measure the antioxidant activities of three natural stilbenes, resveratrol (RSV), ε-viniferin (VNF), and vitisin B (VB), and to compare these activities when these compounds are used in combination (Figure 1). These activities were also compared with those of a standardized stilbene-enriched vine extract (SSVE), obtained from *Vitis vinifera* vine shoots, known for its high antioxidant properties [9]. This extract was characterized and contained, in mass, 33.7% RSV, 63.1% VNF, and 3.2% VB [10]. The antioxidant activities were measured using the FRAP, NO, and DPPH methods, and the interactions between these compounds were obtained using the method of Chou and Talalay using CompuSyn software version 1.0.1 [11].

## 2. Results

### 2.1. Determination of Antioxidant Activities with the DPPH-Scavenging Assay

The DPPH-radical-scavenging antioxidant capacity of RSV, VNF, and VB, individually or in equimolar combinations, are illustrated in Figure 2A and Figure 3. All the molecules and their mixtures showed antioxidant activities at the concentrations used in a dose-dependent manner. RSV had a DPPH-radical-scavenging capacity similar to that of VNF with an IC_50_ of 81.92 ± 9.17 and 80.12 ± 13.79 µM, respectively, while VB was the least active molecule with an IC_50_ of 129.14 ± 26.13 µM (Table 1). All combinations showed additive effects since their CIs were between 0.9 and 1.1 (Table 1).

### 2.2. Determination of Antioxidant Activities with the FRAP Assay

The antioxidant capacity of RSV, VNF, or VB and their equimolar combinations to reduce Fe^3+^ to Fe^2+^ is illustrated in Figure 2B and Figure 4. RSV and VNF showed dose-dependent antioxidant activities at the concentrations used. RSV showed the highest antioxidant capacity, with an IC_50_ of 13.36 ± 0.91 µM, followed by VNF at 28.81 ± 4.15 µM, while the IC_50_ for VB could not be reached at the concentrations tested in this study because of solubility troubleshooting (Table 1). Two combinations, RSV + VNF and RSV + VNF + VB, showed synergistic effects, whereas the combination RSV + VB showed antagonistic effects, and VNF + VB showed additive effects (Table 1).

### 2.3. Determination of Antioxidant Activity with NO-Scavenging Assay

The antioxidant capacity of RSV, VNF, or VB and their equimolar combinations to scavenge the NO radical is illustrated in Figure 2C and Figure 5. All the molecules reduced the amount of NO in a dose-dependent manner at the concentrations used. RSV had the highest NO-scavenging activity, with an IC_50_ of 200.68 ± 15.40 µM, followed by VNF at 338.35 ± 89.47 µM, and finally VB at 368.80 ± 14.20 µM. The RSV + VNF, RSV + VB, and VNF + VB combinations had additive effects, whereas the combination RSV + VNF + VB had synergistic effects (Table 1).

### 2.4. Comparison of the Antioxidant Activities between the Combination RSV + VNF and the Standardized Stilbene-Enriched Vine Extract

The antioxidant activities of the combination RSV + VNF were compared with those of the standardized stilbene-enriched vine extract (SSVE) in different assays (DPPH-, FRAP-, and NO-scavenging) (Figure 6). The SSVE was previously characterized and mainly contained RSV and VNF. The only significant differences were observed at the highest concentrations in the FRAP- and NO-scavenging assays.

## 3. Discussion

Our results showed that RSV and VNF had antioxidant activities in the DPPH, FRAP, and NO assays, whereas VB had antioxidant activities only in the DPPH and NO assays. RSV and VNF showed the highest antioxidant activities. In the literature, the antioxidant potential of RSV has been well documented with DPPH, FRAP, and NO assays. On the other hand, few studies have evaluated the antioxidant potential of VNF and VB, and no studies using FRAP or NO assays have been reported despite their in vitro anti-inflammatory and antioxidant activities in cell cultures [8]. In the literature, the RSV IC_50_ values obtained from the DPPH assay (the most widely used method for measuring its antioxidant activity) have been highly variable (Figure 7). In our study, the RSV IC_50_ (81.92 ± 9.17 µM) was similar to that reported in the study of Ha et al. and the study of Wang et al., who obtained IC_50_ values of 81.2 and 80.5 µM, respectively, but very different from those reported by Joshi et al. (667.18 µM), Tu et al. (285.54 µM), and Lin et al. (24.3 µM) [12,13,14,15,16]. Three studies reported VNF IC_50_ values between 52.6 and 92 µM, which was comparable to the VNF IC_50_ value in our study (80.12 ± 13.79 µM) [9,12,17]. No study was available for VB.

In the FRAP and NO antioxidant assays, RSV was the most active molecule, followed by VNF and then VB, which showed much less activity. With regard to the FRAP assay, it was difficult to compare our results with those in the literature, as they were often expressed in a different manner. Nevertheless, a few studies have calculated IC_50_ values for RSV, in particular, the studies by Lin et al., who obtained an IC_50_ of 20.7 µM, which was very close to the IC_50_ of 15.38 µM that we obtained, but very different from the IC_50_ obtained by Kurin et al. and Skroza et al., which were 162.02 and 335.9 µM, respectively [4,15,18].

Very few studies on NO-scavenging capacity have been carried out with RSV. Man-Ying Chan et al. showed that 50 µM RSV in solution in the presence of ethanol inhibited NO by 46.2%, unlike our study where the inhibition was lower (25.3%) [19].

The antioxidant capacity of these three compounds was evaluated with the ORAC test by Biais et al. [10]. The authors concluded that VNF had the greatest antioxidant capacity, three times greater than the one of RSV and twenty-one times greater than the one of VB. In our study, the antioxidant activity of RSV was higher than that of VNF in two of the three assays used (FRAP and NO). One reason for this difference could have been due to the reaction mechanisms involved. According to Huang et al., two chemical processes shared by the majority of polyphenols were responsible for their antioxidant effectiveness, namely hydrogen atom transfer and electron transfer [20]. The ORAC assay is based on hydrogen atom transfer, whereas the DPPH, FRAP, and NO assays are based on electron transfer.

It is known that the more hydroxyl groups a molecule has, the stronger its antioxidant activity. In our study, RSV was as effective, if not more so, compared with its dimer VNF and tetramer VB, which possess the highest number of hydroxyl groups. This observation could be explained by the fact that RSV, by virtue of its structure, could release its protons more easily than VNF or VB. In the DPPH assay, some authors have stated that the steric accessibility of the DPPH radical is a major determinant of the reaction, so small molecules have better access to the radical site and therefore a higher antioxidant capacity. Conversely, large compounds react slowly, which could explain their lower activity [5,20].

A number of studies have been carried out on the interactions between RSV and other polyphenols. Most of them have shown antagonistic interactions in the DPPH assay. These antagonistic interactions have mainly been shown with polyphenols that belong to the flavonoid and phenolic acid families, such as catechin, quercetin, caffeic acid, kaempferol, and gallic acid [4,14,19,21]. In our study, the stilbenes were able to interact with each other in an additive manner. These results suggested that within the same class of polyphenols, they appeared to cooperate and potentiate their effects, whereas polyphenols that belong to different classes could sometimes exert antagonistic effects. Concerning the FRAP assay, we observed synergistic effects for the RSV + VNF combination, whereas when VB, which is a less active molecule, was combined with RSV, the effect was antagonistic. Here, it could be assumed that VB negatively interacted with RSV by reducing its antioxidant activity. In the literature, Abraham et al. found additive effects between RSV in combination with polyphenols such as chlorogenic acid, pelargonidin, and epigallocatechin gallate [22]. Similarly, Skroza et al. observed synergistic effects when RSV was combined with catechin or caffeic acid [18]. However, when it was combined with gallic acid or quercetin, the interactions led to antagonistic effects. Concerning the NO-scavenging assay, the combination of RSV with VNF or VB provided additive effects. Kurin et al. observed that the interaction of RSV with caffeic acid induced a synergistic effect and the combination with quercetin induced an additive one [4].

To our knowledge, no study has been carried out to measure the interactions that involve VNF or VB, either between these two oligomers or with compounds belonging to other families. We showed that VNF presented interesting antioxidant activities produced in combination with RSV in either additive or synergic interactions, whereas VB presented much less antioxidant activities that could not produce any synergic interaction but even antagonist ones. In the ternary combination, VB reduced the antioxidant activities of the mixture of RSV + VNF, e.g., the IC_50_ for the FRAP assay was 21.61 ± 1.94 µM (RSV + VNF + VB) versus 13.28 ± 1.44 µM (RSV + VNF).

Our results, as well as those reported in the literature, highlighted that the nature of the interactions depended on the mixtures of compounds and the assay used to measure the antioxidant capacity. As an example, Skroza et al. showed that interactions could be highly variable—additive, synergistic, or antagonistic—depending on the compound mixed with RSV (gallic acid, caffeic acid, (+)-catechin, or quercetin) [18]. In addition, Kurin et al. also showed that the combination of RSV and quercetin produced an antagonistic effect in the SRD, FRAP, DPPH, and ABTS assays; an additive effect in the NO-scavenging assay; and a synergistic effect in the NRD and RP assays [4].

Here, we mainly report additive interactions as the calculated combination indexes were close to one. Only the combinations of RSV + VNF and RSV + VNF + VB gave rise to synergistic effects in the FRAP assay, and RSV + VNF + VB in the NO-scavenging assay. Insofar as the calculated combination indexes were between 0.7 and 0.85, we qualified these interactions as weak synergism according to Chou [11]. We also observed an antagonistic interaction between RSV and VB in the FRAP assay that was moderate in nature, according to the same author [11].

The standardized stilbene-enriched vine extract (SSVE), derived from a vine shoot extract, mainly consisted of RSV and VNF, with a very small proportion of VB and unknown compounds. Our results showed that the RSV + VNF combination exhibited similar antioxidant effects compared with those observed for the SSVE in the DPPH-, FRAP-, and NO-scavenging assays. However, at the highest concentrations, the SSVE showed a statistically greater or lesser effect than the combination in the FRAP- and NO-scavenging assays, respectively. These observations showed that the presence of some compounds, even in very low concentrations, could modify the nature of the interactions of the major compounds. Therefore, in order to estimate the nature of compound interactions within a mixture, it would appear necessary to characterize all the compounds present in the extract, even those present in very low concentrations.

Some authors have attempted to explain the mechanisms involved in the interactions. In the case of additive interactions, these authors have assumed that the molecules acted independently of each other. Synergistic or antagonistic interactions, on the other hand, could be explained by a regeneration mechanism: either a less effective antioxidant regenerates the more effective one, which in turn exerts antioxidant activities, hence the phenomenon of synergy, or conversely, the more effective antioxidant regenerates the less effective antioxidant, which in turn exerts a weaker antioxidant activity, hence the phenomenon of antagonism [6,23,24]. In our study, we hypothesized that VB could be regenerated by either RSV or VNF when in combination and therefore produce an antagonistic effect.

**Figure 7 molecules-28-07521-f007:**
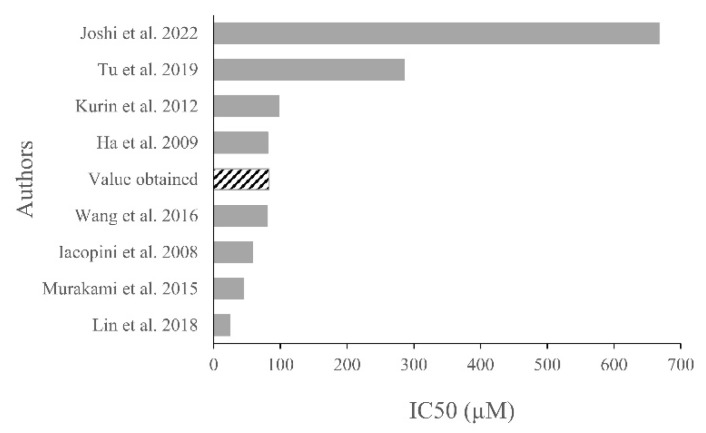
Summary of IC_50_ values for resveratrol in the DPPH assay according to the literature [4,12,13,14,15,16,21,25].

Mixtures of compounds can give rise to interactions that can result in additive, synergistic, or antagonistic effects. It seems clear that additive or synergistic interactions are beneficial in the field of human health insofar as they enable the same or even a greater effect obtained with lower doses of compounds. Conversely, antagonistic interactions are deleterious to the resulting effects. In our case, the mixture of stilbenes essentially produced additive or synergistic interactions. The SSVE we tested, containing a mixture of these stilbenes, therefore tended to have a beneficial effect. This SSVE was simply and rapidly obtained in two steps (an extraction followed by a centrifugal partition chromatography separation step), which represented an advantage for the valorization of vine byproducts and their use for health applications. This last point obviously merits further investigation.

## 4. Materials and Methods

### 4.1. Chemicals

Iron chloride hexahydrate; 2,2-diphenyl 1-picrylhydrazyl (DPPH); sodium nitroprusside (SNP); sodium acetate; 2,4,6-Tri(2-pyridyl)-s-triazine (≥99% (TLC); (TPTZ) and Griess reagent were purchased from Sigma-Aldrich (Saint-Quentin-Fallavier, France). (±)-6-Hydroxy-2,5,7,8-tetramethylchromane-2-carboxylic acid (Trolox) was purchased from Thermo-Fisher (Illkirch, France). Resveratrol (RSV), ε-viniferin (VNF), vitisin B (VB), and standardized stilbene-enriched vine extract (*Vitis vinifera* grapevine shoot extract, named SSVE in the following) were obtained in the laboratory from vine shoots using the method previously described. Briefly, grapevine shoot extract was fractionated using centrifugal partition chromatography with a two-phase solvent system that was composed of heptane, ethyl acetate, methanol, and water (1:2:1:2). The extract sample was injected for a single run that allowed us to collect 7 fractions. These 7 fractions were analyzed with a UHPLC-ESI-MS/MS system to identify the RSV-, VNF- and VB-enriched fractions. This SSVE contained, in mass, 33.7% RSV, 63.1% VNF, and 3.2% VB. The confirmation of the identification of each stilbene was performed with NMR [10].

### 4.2. DPPH-Scavenging Assay

The DPPH assay was slightly modified from that described by Blois [26]. In this test, the purple chromogenic radical DPPH was reduced by antioxidant compounds to a pale-yellow hydrazine compound. The stable DPPH radical (200 µM) was dissolved in methanol. Briefly, RSV, VNF, VB and their equimolar combinations were prepared at different concentrations ranging from 100 to 400 µM. As an example, 100 µM of the combination RSV + VNF contained 50 µM of RSV and 50 µM of VNF. The stilbene-enriched SSVE mainly contained 1/3 RSV and 2/3 VNF, in mass, which represented an equimolar quantity of RSV and VNF. Therefore, the SSVE was prepared at the same concentration as those used for the different combinations of stilbenes, i.e., 100 µM of the SSVE contained 50.9 µM of RSV and 47.9 µM of VNF. The different solutions of stilbenes and their combinations were prepared in a mixture of methanol/water (50/50), and a volume of 50 µL was mixed with 150 µL of DPPH solution in a 96-well plate in order to obtain the final concentration ranging from 25 to 100 µM for an individual compound. The plate was then incubated for 20 min in the dark at 37 °C. Trolox was used as a positive control at concentrations ranging from 50 to 200 µM under the same conditions. DPPH-radical-scavenging activity was estimated by measuring the absorption at 520 nm with a CLARIOstar spectrophotometer (BMG LABTECH, Champigny-sur-Marne, France), reflecting the quantity of DPPH radicals remaining in the solution. The results were expressed as IC_50_, which represented the concentration of the sample required to reduce the DPPH radical amount by 50%.

### 4.3. FRAP Assay

The reducing power of the samples was determined using a FRAP assay described by Benzie and Strain with a few modifications [27]. The FRAP assay, based on the electron transfer mechanism, measures the ability of antioxidants to reduce the ferric complex [Fe (III)-(TPTZ)_2_]^3+^ to the blue ferrous complex [Fe (II)-(TPTZ)_2_]^2+^ in an acid medium. The FRAP reagent was freshly prepared from an acetate buffer (0.3 M, pH = 3.6) and mixed with TPTZ (10 mM in HCL) and iron chloride hexahydrate (20 mM) in a 10:1:1 ratio. The tested samples were dissolved in ethanol/water (50/50). Briefly, RSV, VNF, VB, their equimolar combinations, and the SSVE were prepared at different concentrations ranging from 50 to 600 µM, as described in the section DPPH-scavenging assay. The different solutions of stilbenes and their combinations were prepared in a mixture of methanol/water (50/50), and a volume of 10 µL was mixed with 190 µL of FRAP reagent in a 96-well plate in order to obtain the final concentration ranging from 2.5 to 30 µM for an individual compound. The plate was then incubated for 30 min in the dark at room temperature. Trolox was used at concentrations ranging from 50 to 300 µM as a positive control under the same conditions. The FRAP assay’s reducing power was estimated by measuring absorption at 593 nm using a CLARIOstar spectrophotometer (BMG LABTECH, Champigny-sur-Marne, France). A Trolox concentration of 300 µM was considered 100% of the reduction of the iron complex. The IC_50_ represented the concentrations of the samples required to reduce the iron complex by 50%.

### 4.4. NO-Scavenging Assay

The NO radical was measured using SNP, a compound that spontaneously releases NO in aqueous solutions at a physiological pH under light irradiation. The NO radical interacted with oxygen to generate nitrites, which were then measured using the Griess reagent. Briefly, RSV, VNF, VB, their equimolar combinations, and the SSVE were prepared at different concentrations ranging from 50 to 800 µM, as described in the section DPPH-scavenging assay. Different solutions of stilbenes and their combinations were prepared in a mixture of methanol/water (50/50), and a volume of 200 µL was mixed with 200 µL of SNP (5 mM) in order to obtain the final concentrations ranging from 25 to 400 µM for an individual compound. The tubes were then exposed to a controlled light source (5600 lux tungsten lamp) for 20 min. At the end of the exposure, a 60 µL aliquot of the reaction was sampled, to which 60 µL of Griess reagent was added to 96-well plates. Absorbance was measured after 15 min at 540 nm using a CLARIOstar spectrophotometer (BMG LABTECH, Champigny-sur-Marne, France). The results were expressed as IC_50_, which represented the concentration of the sample required to scavenge 50% of NO production.

### 4.5. Statistical Analysis and Determination of the Interactions

All DPPH-, FRAP-, and NO-scavenging experiments were repeated at least four times, and the results are presented as means ± SEM of a representative experiment. Statistical tests were performed using a one-way ANOVA test followed by a Tukey’s post hoc multiple comparison test. The significance level was set at *p* < 0.05. Interactions were determined using the Chou–Talalay combination index method that quantitatively determined the synergism (CI < 0.9), additivity (0.9 ≤ CI ≤ 1.1), and antagonism (CI > 1.1) of a mixture using CompuSyn software, version 1.0.1 [11].

## 5. Conclusions

This study explored the antioxidant activity of resveratrol, ε-viniferin, and vitisin B, alone or in combination, to highlight their interactions. Resveratrol exerted a greater antioxidant activity, and the interactions between resveratrol and its oligomers resulted in synergistic or additive effects. Because for one given combination, the nature of the interactions depended on the assay, it seemed that the antioxidant activity of the mixtures depended on the reaction mechanisms involved in the assay used. It was therefore essential to use a multiple-approach method to characterize the antioxidant capacity of a compound alone or in a mixture. Finally, the results observed with a grapevine extract confirmed that the presence of some compounds, even in very low amounts, could influence the total activity of a mixture, thus underlining that the prediction of the antioxidant capacity of an extract should be cautious without its exact composition.

## Figures and Tables

**Figure 1 molecules-28-07521-f001:**
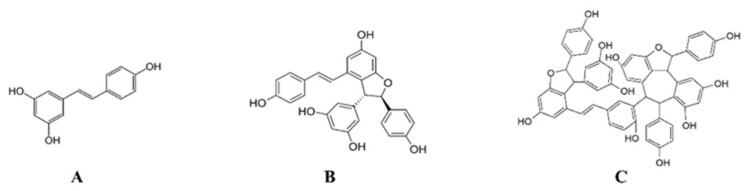
Molecular structures of stilbenes isolated from *Vitis vinifera* vine shoots. (**A**) Resveratrol; (**B**) ε-viniferin; (**C**) vitisin B.

**Figure 2 molecules-28-07521-f002:**
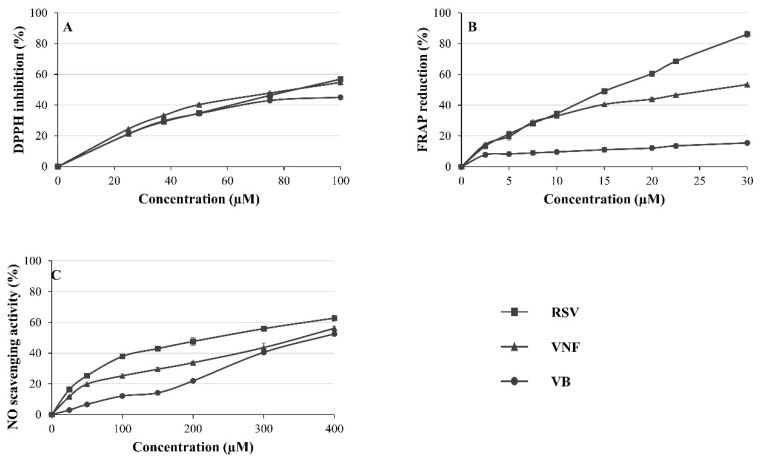
Antioxidant capacity of individual stilbenes (RSV, resveratrol; VNF, ε-viniferin; VB, vitisin B). (**A**) DPPH assay; (**B**) FRAP assay; (**C**) NO-scavenging assay.

**Figure 3 molecules-28-07521-f003:**
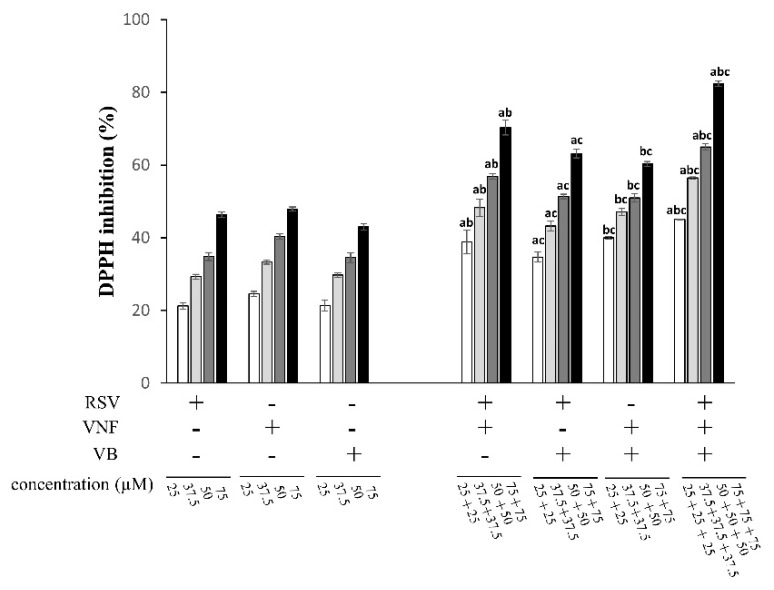
DPPH radical reduction and antioxidant capacity of individual stilbenes (RSV, resveratrol; VNF, ε-viniferin; VB, vitisin B) and their equimolar combinations. The 100% marker represents the DPPH alone, which was used as a control. The result of the representative experiment is shown. Values are expressed as mean ± standard deviation (*n* = 3). Data were analyzed with an ANOVA (*p* < 0.05). Each letter (a, b, and c) indicates that the combinations were statistically different from the individual compounds (a for resveratrol, b for VNF, and c for VB) according to a post hoc Tukey comparison test at *p* < 0.05.

**Figure 4 molecules-28-07521-f004:**
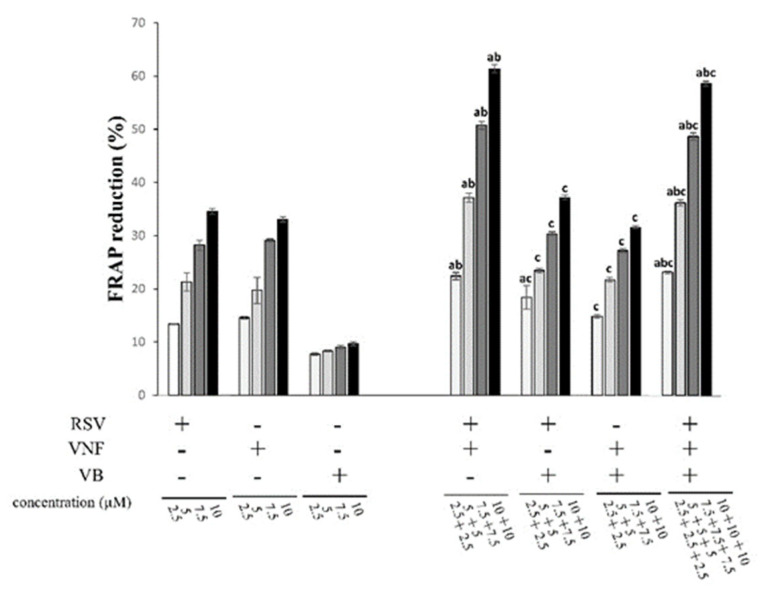
Ferric iron reduction capacity of individual stilbenes (RSV, resveratrol; VNF, ε-viniferin; VB, vitisin B) and their equimolar combinations. The 100% marker represents the Trolox that was used as a positive control. The result of the representative experiment is shown. Values are expressed as mean ± standard deviation (*n* = 4). Data were analyzed with an ANOVA (*p* < 0.05). Each letter (a, b, and c) indicates that the combinations were statistically different from the individual compounds (a for resveratrol, b for VNF, and c for VB) according to a post hoc Tukey comparison test at *p* < 0.05.

**Figure 5 molecules-28-07521-f005:**
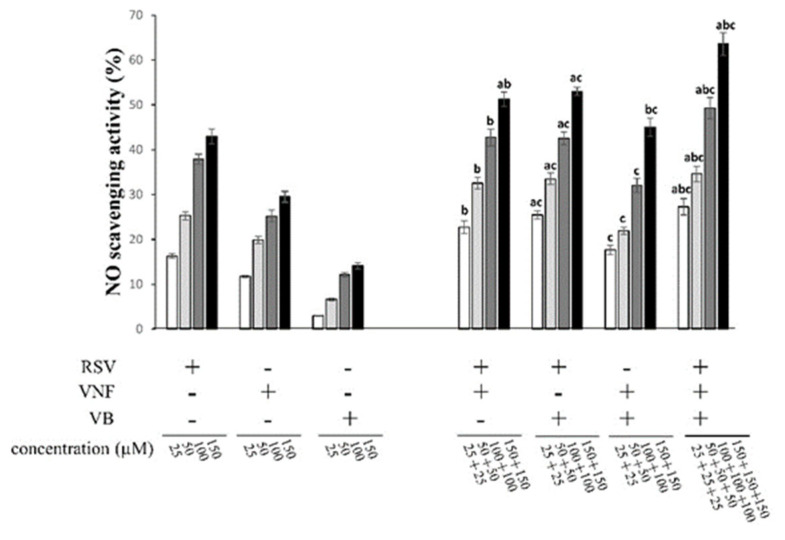
NO-scavenging capacity of individual stilbenes (RSV, resveratrol; VNF, ε-viniferin; VB, vitisin B) and their equimolar combinations. The 100% marker represents the SNP alone, which was used as a positive control. The result of the representative experiment is illustrated. Values are expressed as mean ± standard deviation (*n* = 4). Data were analyzed with an ANOVA (*p* < 0.05). Each letter (a, b, and c) indicates that the combinations were statistically different from the individual compounds (a for resveratrol, b for VNF, and c for VB) according to a post hoc Tukey comparison test at *p* < 0.05.

**Figure 6 molecules-28-07521-f006:**
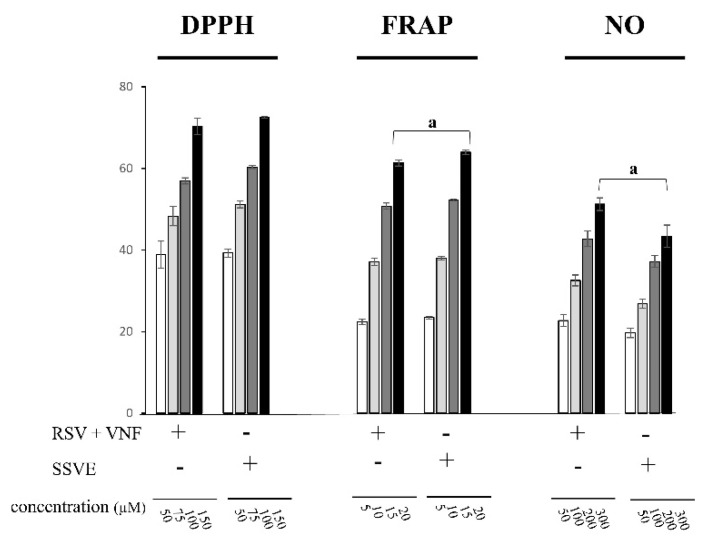
Comparison of the antioxidant activities between the combination (RSV + VNF) and the standardized stilbene-enriched vine extract (SSVE). Data were analyzed with an ANOVA (*p* < 0.05). The letter a indicates that the combination was statistically different from the SSVE according to a post hoc Tukey comparison test at *p* < 0.05.

**Table 1 molecules-28-07521-t001:** CIs of the antioxidant capacities of grapevine stilbenes and their interactions using the DPPH-, FRAP-, and NO-scavenging methods.

	DPPH			FRAP			NO		
	IC50 (µM)	CI	Interaction	IC50 (µM)	CI	Interaction	IC50 (µM)	CI	Interaction
RSV	81.92 ± 9.17			13.36 ± 0.91			200.68 ± 15.40		
VNF	80.12 ± 13.79			28.81 ± 4.15			338.35 ± 89.47		
VB	129.14 ± 26.13			ND			368.80 ± 14.20		
RSV + VNF	71.55 ± 7.26	0.89 ± 0.06	Ad	13.28 ± 1.44	0.70 ± 0.12	Sy	259.78 ± 40.56	1.03 ± 0.01	Ad
RSV + VB	87.82 ± 19.05	0.88 ± 0.06	Ad	ND	1.36 ± 0.34	An	243.07 ± 52.03	0.87 ± 0.13	Ad
VNF + VB	89.07 ± 11.41	0.91 ± 0.05	Ad	ND	0.96 ± 0.20	Ad	ND	0.99 ± 0.17	Ad
RSV + VNF + VB	90.42 ± 10.99	0.98 ± 0.04	Ad	21.61 ± 1.94	0.81 ± 0.06	Sy	216.24 ± 80.03	0.72 ± 0.18	Sy

CI, combination index; Ad, additive; Sy, synergy; An, antagonism.

## Data Availability

The data presented in this study are available on request from the corresponding author.

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
