# Peer review of "Resveratrol, ε-Viniferin, and Vitisin B from Vine: Comparison of Their In Vitro Antioxidant Activities and Study of Their Interactions"

_molecules, 2023, doi:10.3390/molecules28227521_

Round 1

Reviewer 1 Report

Comments and Suggestions for Authors

This manuscript aimed to compare in vitro antioxidant activities of three stilbenes and to explore their interactions in exerting antioxidant effect (radical scavenging in the first place). The authors used three different antioxidant assays and Chou and Talalay method to investigate potential interactions. Overall, this simple study is well designed, methods are generally appropriate, and the obtained results bring some novelty to this research field. However, there are several concerns in the manuscript that need to be clarified.

Since all the assays were done in triplicate, combination index (CI) should be calculated for three independent experiments to obtain three CI values with the mean and standard deviation. Besides, CI in a range 0.8 – 1.2 usually indicates an additive effect (I doubt that CI of 1.11 indicates antagonism). In my opinion, only the combination RSV + VB had the real antagonistic effect. Anyway, the values should be discussed too, not only existence/absence of interactions.

The order of results should be a little bit modified – Table 1 should be before Figure 3 since it contains data for all three antioxidant assays.

The composition of stilbene-enriched extract obtained from Vitis vinifera vine shoots should be stated in the Methods instead of Introduction. Besides, the term ‘fraction’ itself should be avoided in the text in my opinion and replaced with standardized stilbene-enriched vine extract or something similar.

Some other concerns include:

Check the order of name and surname in the list of authors.

Line 29: the symbol for radical should be modified (it should be dot instead of a circle).

Line 95: ‘least’ instead of ‘less’

Line 117: refer to table 1 and not just table.

Line 155: The first sentence is not completely correct since some stilbenes did not show antioxidant activities in some assays. Please modify it.

Figure 7 is not necessary in my opinion. Besides, it should be IC50 instead of CI50.

Comments on the Quality of English Language

No major issues were detected.

Author Response

Dear reviewers,

Please find as attached file point by point responses to your comments. In the revised version of our paper, we hilihgthed in red the changes we made.

We hope we provided the additionnal informations you requested.

Best regards

Arnaud Courtois

Reviewer 2 Report

Comments and Suggestions for Authors

The article evaluates the antioxidant capacities of three grapevine stilbenes - RSV (resveratrol), VNF (ε-viniferin), and VB (vitisin B) - both individually and in equimolar combinations. The antioxidant activities were measured using three different assays: DPPH scavenging, FRAP assay, and NO scavenging assay. The study provides a comprehensive analysis, comparing individual activities of these compounds and their combinations.

       1.            The statement about the antioxidant synergy between vitamins C and E, while interesting, could be more directly related to the main topic of the article. If this example is to be included, it might be beneficial to explain its relevance to the study of polyphenols, especially stilbenes.

       2.            It's great that the aim of the study is explicitly mentioned. However, for a clearer understanding of its significance, a brief background on why studying these specific stilbenes (RSV, VNF, and VB) is important could be added.

       3.            You need to put reference at line 83-84. “This extract was characterized and contains, in mass, 83 33.7% RSV and 63.1% VNF, 3.2% VB.”

       4.            Some redundancy is observed in the descriptions; for example, stating that molecules were tested "individually or in equimolar combinations" is mentioned repeatedly. A more concise presentation might be beneficial for the reader.

       5.            The report states "the IC50 for VB could not be reached at the concentrations tested in this study." It may be valuable to specify why higher concentrations were not tested or if there are any potential implications of this finding.

       6.            When discussing interaction types (additive, synergistic, or antagonistic), consider using visual aids or diagrams to help readers visualize these concepts.

       7.            In 4.1 Chemicals there is a reference not numbered but mentioned (Biais et 261 al. 2017). Even if you put the references some details about the methods must be put about how were they obtained.

       8.            The study is irrelevant if there is no characterization of resveratrol (RSV), ε-viniferin (VNF), vitisin B (VB) and fraction 5 obtained in the laboratory.

       9.            The DPPH assay mentions "The results were expressed as IC50, which represent the concentration of the sample required to reduce the DPPH radicals amount by 50%." This is a clear definition. However, in subsequent assays, the definition of IC50 is varied. It would be clearer to maintain a single, consistent definition.

   10.            The statistical methods section could benefit from a bit more elaboration. For instance, mentioning whether the ANOVA is one-way or two-way and detailing any assumptions or preprocessing done before running the test might be helpful for replicability.

   11.            It's good that CompuSyn software is mentioned in the context of the Chou-Talalay method. However, it might be useful to add software version and any other parameters used.

   12.            The paper occasionally offers figures for comparisons (like the IC50 values) but at other times suggests there are discrepancies in the literature without giving exact numbers.

   13.            The findings might benefit from a more detailed discussion section, exploring the implications and potential applications.

Comments on the Quality of English Language

       1.            Some sentences could be made more concise without losing their meaning. For example, "Numerous analytical methods exist in order to measure the antioxidant capacity" could be simplified to "Numerous methods measure the antioxidant capacity."

Author Response

Dear reviewers,

Please find as attached file our point by point responses to your comments. We have hillighthed in red the changes we have made in our article.

We hope we provided all the additionnal informations you requested.

Best regards

Arnaud Courtois

Round 2

Reviewer 1 Report

Comments and Suggestions for Authors

The authors improved their initial version of the manuscript. I support it to be published in this form.

Comments on the Quality of English Language

No major issues were detected.

Reviewer 2 Report

Comments and Suggestions for Authors

I noted that the changes made have addressed my previous comments and suggestions. The article is good for publication.